# Clinical Insights into Structure, Regulation, and Targeting of ABL Kinases in Human Leukemia

**DOI:** 10.3390/ijms25063307

**Published:** 2024-03-14

**Authors:** Andrew Wu, Xiaohu Liu, Clark Fruhstorfer, Xiaoyan Jiang

**Affiliations:** 1Collings Stevens Chronic Leukemia Research Laboratory, Terry Fox Laboratory, British Columbia Cancer Research Institute, Vancouver, BC V5Z 1L3, Canada; awu@bccrc.ca (A.W.); wliu@virogin.com (X.L.);; 2Department of Medicine, University of British Columbia, Vancouver, BC V6T 1Z4, Canada; 3Department of Medical Genetics, University of British Columbia, Vancouver, BC V6T 1Z4, Canada

**Keywords:** BCR::ABL1, ABL regulation, X-ray crystallographic structures, chronic myeloid leukemia, leukemia stem cell, small-molecule inhibitors, tyrosine kinase inhibitors, TKI resistance, BCR::ABL1 mutations

## Abstract

Chronic myeloid leukemia is a multistep, multi-lineage myeloproliferative disease that originates from a translocation event between chromosome 9 and chromosome 22 within the hematopoietic stem cell compartment. The resultant fusion protein BCR::ABL1 is a constitutively active tyrosine kinase that can phosphorylate multiple downstream signaling molecules to promote cellular survival and inhibit apoptosis. Currently, tyrosine kinase inhibitors (TKIs), which impair ABL1 kinase activity by preventing ATP entry, are widely used as a successful therapeutic in CML treatment. However, disease relapses and the emergence of resistant clones have become a critical issue for CML therapeutics. Two main reasons behind the persisting obstacles to treatment are the acquired mutations in the ABL1 kinase domain and the presence of quiescent CML leukemia stem cells (LSCs) in the bone marrow, both of which can confer resistance to TKI therapy. In this article, we systemically review the structural and molecular properties of the critical domains of BCR::ABL1 and how understanding the essential role of BCR::ABL1 kinase activity has provided a solid foundation for the successful development of molecularly targeted therapy in CML. Comparison of responses and resistance to multiple BCR::ABL1 TKIs in clinical studies and current combination treatment strategies are also extensively discussed in this article.

## 1. Introduction

Chronic myeloid leukemia (CML) is characterized by the unregulated proliferation of myeloid granulocytic cells originating from leukemic stem cells (LSCs) and myeloid progenitors in the bone marrow. As of 2023, the disease accounts for about 15% of newly diagnosed leukemias with a median age of 60 years [1]. The primary defining feature of CML for diagnosis is the presence of the BCR::ABL1 chimeric fusion oncogene, or Philadelphia chromosome (Ph), which arises from a t(9;22)(q34;q11) translocation between chromosomes 9 and 22 [2]. Based on the European LeukemiaNet (ELN) and MD Anderson (MDACC) criteria, CML progresses through three stages: chronic (CP), accelerated phase (AP), and blast crisis (BC), with each phase defined by increasing blast percentages, accumulation of additional cytogenic abnormalities, and other hematologic parameters [3,4]. However, since 2022, the World Health Organization (WHO) has removed the CML-AP designation and instead stratifies CML into CML-CP and CML-BP only, based on the presence of high-risk features [5]. The precise classification of CML continues to be a matter of debate.

The p210 isoform of the BCR::ABL1 oncogene is the main driver of CML pathogenesis in the majority of CML patients [6]. On the other hand, the p190 isoform is predominantly expressed in Ph^+^ acute lymphoblastic leukemia (ALL) and is associated with a worse prognosis [7]. Interestingly, the p230 isoform was found to comprise a distinct clinical entity called neutrophilic CML, with a more benign clinical course than that associated with the p210 and p190 isoforms [8]. After translocation, the coiled-coil domain of BCR::ABL1 and the removal of specific regulatory structures like the N-terminal myristoyl and Cap region allow for constitutive kinase activity. These isoforms have been known to promote various oncogenic signaling pathways which increase cell proliferation, survival, and erroneous DNA repair mechanisms. Because CML pathogenesis is heavily dependent on the p210 isoform of the BCR::ABL1, this protein also serves as an attractive target for therapies such as tyrosine kinase inhibitors (TKI). Indeed, TKIs have demonstrated remarkable efficacy in managing the disease and even extending the life expectancy of CML patients to near-normal levels, and the goals for managing CML have shifted from survival to treatment-free remission (TFR) [9]. Despite the success of TKIs, they are rarely curative, and patients are often required to adhere to lifelong treatment which accumulates economic burden. Although TFR and TKI discontinuation is achievable for CML patients, there is a lack of consistency in TFR treatment policies, and in many cases, treatment discontinuation can lead to eventual disease relapse due to the persistence of CML cells with BCR::ABL1 mutations or LSCs that harbor various BCR::ABL1-independent resistance mechanisms [10,11,12,13,14,15,16,17].

In this review, we provide an up-to-date structural characterization of the ABL1 kinases and further describe their complexes with past, present, and future inhibitors. Specifically, we focus on the ABL1B isoform, referred to as ABL1, and compare its domain structure to that of BCR::ABL1. We also review several elegant clinical studies investigating the clinical response and resistance of CML patients to six FDA-approved TKIs, TKI dose optimization strategies, and the efficacy of combination treatment strategies as compared to TKI monotherapies to improve treatment outcomes and overcome drug resistance.

## 2. Structural and Molecular Description of ABL1 Regulation

In contrast to the highly regulated wild-type ABL1 kinase, BCR::ABL1 escapes spatial and temporal control to be constitutively activated (Figure 1) [18]. To better understand ABL1 regulation, we discuss the different conformations of the ABL1 kinase domain, the mediation of ABL1 autoinhibition via regulatory domains, and the consequences of losing autoinhibition after BCR fusion.

### 2.1. ABL1 Kinase Domain

The ABL1 kinase domain is a bi-lobal structure with a myristoylated NH_2_-terminal (N-lobe) and the COOH-terminal (C-lobe, Figure 1 and Figure 2) [18,19]. The N-lobe includes the αC helix and phosphate binding loop (P-loop), whereas the C-lobe houses the peptide substrate-binding site [20]. In between the N-lobe and the C-lobe is the active site which comprises the ATP-binding pocket and activation loop (A-loop, Figure 1 and Figure 2A) [20,21].

Like other kinases, ABL1 switches between active and inactive conformations. In the active state, the A-loop is in an open conformation and provides a platform for substrate binding (Figure 2A) [22,23]. The phosphorylation of Y393 in the A-loop stabilizes the active conformation by hydrogen bonding with R386 and forming salt bridges with R362 and R384 [24,25,26]. One of the hallmark structural features of the active state is the adoption of the “DFG-in” motif (D381-F382-G383) at the N-terminal base of the A-loop [26]. In this activated state, D381 points towards the ATP-binding site and coordinates a Mg^2+^ ion for catalysis, while another residue, F382, is buried in a hydrophobic pocket (Figure 2B) [26]. Concurrently, the substrate-binding site at the C-terminus accommodates substrate phosphorylation and downstream signaling [26].

In contrast, the inactive form of ABL1 has a “DFG-out” motif where the activation loop folds back into the ATP-binding site to block Mg^2+^ from initiating catalysis (Figure 2B) [26]. This causes the unphosphorylated Y393 residue to replace kinase-targeting substrates in the active site and further frees the R386 residue to move toward E282 under the αC-helix [26]. Meanwhile, the DFG motif undergoes a drastic rotation such that the F382 residue occupies the ATP-binding site and contacts other residues (Figure 2B) [26]. This stabilizes the inactive conformation and creates a specificity pocket behind the gatekeeper residue T315 which increases the binding specificity of several TKIs [27,28].

In addition, there is a third intermediate conformation of the ABL1 kinase domain which resembles c-Src in the inactive state, hence its name ‘Src-like inactive ABL1’ (Figure 2C) [29]. Like the c-Src inactive conformation, the A-loop is present in a compacted form that blocks the substrate binding site [28,29,30]. However, unlike the regular inactive ABL1, the Src-like inactive ABL1 exists with the DFG-in motif [29]. Furthermore, to relieve the strong steric hinderance in the active site, the αC-helix is swung out of the active site, creating an enlarged ATP-binding pocket that is believed to facilitate the flipping of the DFG motif during the inactive/active conformation switch [29]. The major structural change from adopting the Src-like inactive state may accommodate kinase domain mutations that promote ABL1 kinase activity while facilitating escape from TKI targeting [29]. The discovery of this intermediate state has thus encouraged the development of alternative strategies to design new ABL1 inhibitors that can overcome this structural change in the kinase domain [29].

### 2.2. ABL SH2 and SH3 Domains

The SH2 and SH3 domains of ABL1 play a vital role in ABL autoinhibition. In the inhibited form of ABL1, the SH2 and SH3 domains function as a clamp to restrict the conformation of the kinase domain to remain in the inactive form (Figure 2D) [19,31]. Specifically, the SH2 domain physically docks onto the distal side of the C-lobe through an extensive network of hydrophobic interactions and hydrogen bonding such that the movement of the kinase C-lobe is severely restrained (Figure 2D) [31,32]. In contrast, the SH3 domain forms interactions with the kinase N-lobe, which also maintains the ABL1 autoinhibition (Figure 2D) [31,32]. The SH3 domain also forms many intermolecular interactions with the SH2-kinase adjoining linker region, which functions as the cognate ligand for SH3 domains. In addition, the connector between the SH2 and SH3 domains adopts a rigid structure that further stabilizes these domains into the kinase domain, hence reinforcing the inhibitory effect (Figure 2D) [19,31,32]. Thus, for ABL1 to be activated, the rigid SH2-SH3 clamp must be removed (Figure 2D). In fact, biochemical and crystallographic studies have shown that upon activation, the SH2 domain dissociates from the C-lobe and repositions onto the N-lobe, thereby diminishing autoinhibition and allowing ABL1 to adopt an elongated conformation [31]. This activated conformation is proposed to result from two possibilities. Firstly, the phosphorylation of the SH2-kinase linker located Tyr-245 residue disrupts the binding of the SH3 domain [19]. Another possibility is for the SH2 domain to bind to the phosphorylated tyrosine of ABL1 substrates, resulting in the dissociation of the SH2 domain from the C-lobe [19]. In both cases, the rigid SH2-SH3 clamp is removed from ABL1 which allows for SH2- or SH3- processive phosphorylation of substrates.

### 2.3. ABL1 N-Terminal Myristoylation Group and Cap Region

The N-terminal myristoylation group and Cap region play a crucial role in ABL1 regulation, as they act as another autoinhibitory mechanism by binding to the myristoyl pocket (Figure 2D) [33]. Instead of functioning as an anchor for membrane binding, the N-terminally located myristoylation group is bound in a deep hydrophobic pocket at the base of the C-lobe of the ABL1 kinase domain [19]. Insertion of this myristoylation group results in a 90° bend in the middle region of the C-terminal-located α1 helix, which otherwise exists in an extended form and occludes the SH2 domain from adopting onto the C-lobe for inhibitory machinery assembly [32]. The Cap region, which connects the myristoylation group and SH3 domain, positions the myristoylation group for binding to the C-lobe. Moreover, it facilitates SH2-SH3 unit fastening to the rear of the ABL1 kinase domain by making extensive contact with the SH2 domain, the SH2-SH3 connector, and possibly the SH3 domain [31]. In the BCR::ABL1 fusion protein, the N-terminal myristoylation group and Cap region are removed after translocation, resulting in an open myristoyl binding pocket and uncontrolled ABL kinase activity. Thus, allosteric inhibitors that target the myristoyl binding pocket have also been successful in the treatment of CML [34,35,36].

### 2.4. BCR::ABL1 Coiled-Coil Domain

In addition, the BCR::ABL1 translocation also induces the gain of the coiled-coil domain at the BCR N-terminal, severely abrogates the ABL1 autoregulation, and changes the kinase dynamics, resulting in subsequent phosphorylation of numerous substrates, including GRB2/GAB2, CRKL, JAK/STAT, MAPK, and PI3K/AKT pathways (Figure 2E) [36,37,38,39]. In particular, this coiled-coil domain is responsible for BCR::ABL1 tetramerization, which increases the chances of ABL activation by proximity-induced trans-phosphorylation of Y393 in the A-loop and constitutive activation [37]. Hence, the oligomeric state of BCR::ABL1 is essential for its high kinase activity and transforming ability. Moreover, it has been shown that coiled-coil-deleted BCR::ABL1 fails to induce CML-like disease in mouse models, indicating the vital role of this domain in BCR::ABL1 transformation [40].

## 3. Structural Characterization of BCR::ABL1 Kinase Inhibitors

Understanding molecular and X-ray crystallographic structures of BCR::ABL1 oncoprotein has led to the development of several potent TKIs for the treatment of CML in the clinical [36]. Currently, six BCR::ABL1-targeting TKIs have been approved for CML frontline therapy. These include the first-generation TKI Imatinib (IM), second-generation TKIs Nilotinib (NL), Dasatinib (DA), and Bosutinib (BOS), the third-generation TKIs Ponatinib (PON), and most recently, the allosteric TKI Asciminib (ASC). These TKIs have been extensively characterized and are distinguished by their binding mechanism to BCR::ABL1 (Figure 3).

### 3.1. BCR::ABL1 DFG-Out TKIs

#### 3.1.1. Imatinib

Briefly, IM, NL, and PON are TKIs that bind specifically to the ABL1 kinase domain in the DFG-out (inactive) conformation (Figure 3A). In the ABL1-IM complex, IM locks ABL1 in the inactive conformation by inserting into the central region of the kinase domain between the αC helix and A-loop (Figure 4A) [41]. The interaction is stabilized by six hydrogen bonds and numerous van der Waals interactions between IM with the surrounding residues of the active site. Consequently, the association of IM and the ABL1 kinase domain relies on the adoption of ABL1 into the inactive conformation and the integrity of the total interaction network. Hence, kinase domain mutations, which cause loss of drug interaction contacts, destabilization of the inactive conformation (disruption of the kinked P-loop conformation), or an active-conformation favoring equilibrium shift, result in decreased or even total loss of binding between IM and the ABL1 kinase [42]. The well-documented T315I mutation provides an excellent example of how a single mutation greatly impairs IM binding (Figure 3C). The substitution of isoleucine leads to the loss of hydrogen bonding between the T315 residue and IM, which increases active site steric hindrance, thereby restricting access to the ATP-binding site. Moreover, this event reinforces the “activating” hydrophobic spine formation, which results in inefficient IM binding and loss of its therapeutic effects [42,43].

#### 3.1.2. Nilotinib

NL is 20–50-fold more potent than IM, due to more extensive van der Waals interactions between NL and the ABL1 kinase domain which increases binding affinity (Figure 3A and Figure 4B) [34]. This creates a better fit in the binding cleft and limits solvent exposure such that the energy required for desolvation is conserved, unlike for the IM binding [45]. While studies have shown that NL can overcome several IM-resistant BCR::ABL1 mutations, NL is still susceptible to the T315I mutation as well as other compound mutations [46,47].

#### 3.1.3. Ponatinib

PON is a third-generation TKI that is 500 times more potent than IM and shows efficacy against the T315I mutation. Although the PON-ABL complex resembles that of IM and NL, there are several key differences (Figure 3A and Figure 4C) [48]. Firstly, the key structural attribute of PON is the ethynyl linker that accommodates the T315/T315I mutation with favorable van der Walls interactions instead of hydrogen bonding which effectively eliminates steric hinderance issues [49]. Secondly, a functional group of PON binds to the pocket vacated by F382 in the DFG-out mode to further increase the binding affinity [49]. Despite high selectivity against nearly all kinase mutations, in vitro binding assays showed that PON is still vulnerable to BCR::ABL1 P-loop mutations, such as E255V, which appears to be a common issue for TKIs that bind to the inactive ABL1 conformation as the integrity of the P-loop is essential for maintaining the inactive ABL1 conformation and binding affinity of DFG-out TKIs [48,49].

### 3.2. BCR::ABL1 DFG-In TKIs

#### 3.2.1. Dasatinib

DA is an oral multityrosine BCR::ABL1 and SRC family tyrosine inhibitor that is 300-fold more potent than IM and can overcome most of the IM-resistant mutations except T315I [50,51]. Unlike the previous TKIs discussed, DA binds to the ABL DFG-in (active) conformation and has fewer interactions with the ABL1 kinase domain (Figure 3B and Figure 4A) [24,41]. Firstly, the functional group on the right end of DA does not protrude into the specific pocket, but rather occupies a hydrophobic pocket created by T315, V299, and three other surrounding residues; hence, the DFG motif no longer needs to be fixed to any certain conformation to facilitate the inhibitor binding (Figure 4A). This allows the A-loop to adopt the extended form in the activated state [24]. Secondly, instead of lying deep in the binding cleft, DA extends in the opposite direction such that the leftmost group forms two hydrogen bonds and numerous Van Der Waals interactions with the hinge region [24]. In addition, in the DA-ABL complex, the P-loop seems to possess mobility, as it is not required to be in a special conformation (Figure 4A) [41]. This suggests that a stringent P-loop conformation is not necessary for the DA binding. Therefore, the flexibility of the DA-binding mode might grant DA the ability to target both the active and inactive conformations of the ABL1 kinase, which could account for the high affinity and efficacy of the drug [24]. However, this idea was challenged by an NMR study, which showed DA primarily interacts with ABL1 in its active state [52]. Additionally, the high degree of P-loop freedom may indicate that no critical contacts are made between DA and the P-loop. This may partially explain the high activity of DA towards the P-loop mutations. Despite the increased sensitivity to a broad spectrum of ABL1 mutants, DA exhibits much less inhibitory effects against the mutations around two essential binding sites: the hinge region and hydrophobic pocket. The hydrogen bonding and van der Waals contact between T315 and DA make the T315 residue indispensable for DA binding [24]. Thus, even the less common T315A mutant shows much higher resistance to DA than NL or PON. Compared to DA, NL and PON form more contacts with the ABL inactive conformation such that the loss of some contacts can be compensated by others [53]. Two clinically resistant mutations against DA—F317L in the hinge region and V299L—exemplify this reasoning well.

#### 3.2.2. Bosutinib

Like DA, BOS is also a dual ABL1-Src kinase inhibitor that binds to the ABL1 kinase domain in its active conformation (Figure 3B and Figure 4D) [54]. However, in the BOS-ABL1 complex, the DFG motif adopts a DFG-flip conformation [23,54]. Specifically, the D381 and F382 residues in the DFG-flip conformation occupy the space where F382 and D381 reside in the DFG-in conformation, respectively. Other than this feature, BOS and DA bind to ABL1 in a very similar pattern, and both share similar drug potency and mutation selectivity (Figure 3). One notable structural distinction of BOS is that it forms multiple crucial van der Waals interactions with the T315 residue in contrast to DA, which usually forms hydrogen bonding with the T315 residue [54]. However, the disruption of this interaction resulting from a T315I mutation can still result in the BOS dissociation from ABL1. Another clinically relevant mutation is V299L, which is located in the hydrophobic binding pocket around T315. The leucine substitute results in steric clashes with BOS, thereby reducing the drug-binding affinity [46,54].

### 3.3. Allosteric Inhibitors of the ABL1 Kinase Domain

#### Asciminib

Despite the efficacy of competitive TKIs described above, kinase mutations (e.g., T315I) that hinder drug binding remain major obstacles to overcome. This has led to the development of non-competitive allosteric inhibitors of BCR::ABL1, such as ASC, that specifically target the ABL myristoyl pocket (STAMP) [34,55,56]. ASC mimics the N-terminal myristoylation group that is lost as a result of BCR::ABL1 fusion and inserts in the myristate binding pocket which redocks the SH2 domain to the kinase domain through the α1 helix bending and induces a conformational change to the BCR::ABL1 inactive state (Figure 3 and Figure 4B) [34,57]. Unlike conventional TKIs like DA and BOS that can target the kinase domain of other tyrosine kinases such as SRC, ASC has been shown to have fewer off-target effects as there are fewer tyrosine kinases with myristoyl binding sites. However, point mutations in BCR::ABL1 affecting the myristoyl pocket such as the A337V, P465S, V468F, and C464W sites have been reported to generate resistance that hinders ASC binding [58]. Additionally, ASC resistance has been found to arise from upregulated ABCG2-mediated drug efflux mechanisms [58]. To overcome these ASC resistance mechanisms, studies are evaluating the combination of ASC with conventional TKIs like PON, which can exert inhibitory effects on ABCG2, and demonstrated success in eliminating T315I mutated CML primary patient cells [59,60].

### 3.4. Other BCR::ABL1 Inhibitors

#### 3.4.1. Axitinib

Two other TKIs, axitinib and rebastinib, were also considered for CML therapies as they both could target the T315I mutation. Axitinib is a vascular endothelial growth factor receptor (VEGFR) inhibitor that was found to selectively inhibit the T315I-mutated BCR::ABL1 in CML and ALL through structural profiling and ex vivo characterization of drug sensitivity and resistance in patient cells [61]. Mechanistically, axitinib binds T315I-mutated BCR::ABL1 in the A-loop DFG-in (active) conformation and prevents kinase autophosphorylation (Figure 4E) [61]. While axitinib was able to demonstrate significant anti-leukemic activity against T315I-mutated cells in vitro, certain drawbacks prevented its adoption in the clinic [61,62]. These included the observations that axitinib was unable to target wild-type BCR::ABL1 and was ineffective against cells with compound mutations [63]. Furthermore, doses required for the efficacy of axitinib monotherapy were not tolerable in patients and did not display added benefit even when combined with other TKIs like ASC [63]. A clinical trial that aimed to compare the efficacy of axitinib vs. BOS by the M.D. Anderson Cancer Center was also terminated due to low accrual of patients due to competing studies.

#### 3.4.2. Rebastinib

Rebastinib is a non-competitive BCR::ABL1 conformation switch control inhibitor that also demonstrated efficacy in targeting T315I-mutated CML cells [20]. When bound, hydrophobic interactions between rebastinib and BCR::ABL1 force the DFG motif away from the catalytic site, which prevents BCR::ABL1 autophosphorylation (Figure 4F) [20,64]. In both in vitro studies and a phase I clinical trial, rebastinib performed well in exerting anti-leukemic effects and inhibiting downstream substrates of BCR::ABL1 [65,66]. However, the clinical benefit observed was deemed to be insufficient for further development of rebastinib in the CML therapeutic pipeline [66]. Despite both axitinib and rebastinib no longer being actively developed, the characterization of these inhibitors provides value in the discovery of future BCR::ABL1 inhibitors.

## 4. TKI Treatment Strategies in the Clinic

Since the advent of IM, the different generations of TKIs have been successful in managing chronic phase CML and extending the life expectancy of patients to near-normal levels. Thus, the focus in CML therapy has now shifted to emphasize the achievement of treatment discontinuation and treatment-free remission (TFR) in patients [67]. However, some patients in the accelerated phase or blast crisis still fail treatment and can progress into sudden blast crisis [68,69,70]. Thus, there is a continuing need to develop curative treatments and to predict patient responses to specific modalities, as well as to anticipate disease relapse and/or progression. These concerns have renewed interest in developing new treatment strategies to overcome TKI tolerance and resistance.

### 4.1. Long-Term Clinical Efficacy of TKIs and TKI Tolerance

To be considered for treatment discontinuation, patients must first experience a sustained deep molecular response (DMR, ≤0.01 BCR::ABL1^IS^), and it has been shown that those who discontinue treatment after a longer duration of DMR have a higher TFR rate [67,71,72,73]. Overall, TFR rates are estimated to range between 25 and 55%, and different treatment strategies such as rotating TKIs are often employed to chase a TFR status for patients who do not experience a DMR [9,67,71,72,73]. However, the extent of success with these strategies remains debatable as some patients who switch TKIs do not experience any meaningful outcomes [1]. Historically, patients have also undergone TKI switching when experiencing TKI toxicity when it is now clear that lower dose scheduling is preferred. For example, DA at 50 mg QD was found to be just as effective and safer than the original 100 mg QD regimen [74,75,76]. Lower dosing of TKIs is not only less toxic but also more affordable for patients.

Lack of TKI affordability is a major contributor to treatment non-compliance and treatment failure. Patients who do not end up achieving TFR are often required to remain on lifelong TKI therapy due to the high incidence of relapse upon treatment discontinuation, which increases the economic burden on patients, and thus, despite the efficacy of next-generation TKIs, they are often inaccessible to patients due to lack of affordability. For comparison, the cost of patented TKIs can be over 250,000 USD per year whereas IM, which is the only TKI currently available as a generic formulation, is drastically more affordable ranging from US $300 to $3000 per year [1,9]. As a result, CML patients are limited to TKIs that provide the most optimal cost-benefit.

Currently, treatment strategies start with the selection of a front-line TKI, which is based on several factors such as efficacy, cost, and toxicity [9]. Furthermore, increasing studies are uncovering endogenous biomarkers that can indicate TKI resistance and can predict treatment response which may also influence front-line TKI selection. In particular, there is a growing interest in identifying unique biomarkers that can predict TKI response at diagnosis, which would allow clinicians to proactively optimize treatment plans for patients. For example, it has been reported that white blood count (WBC) can predict deep molecular responses to TKI in newly diagnosed CML patients, since patients with low WBC at diagnosis were more likely to be TKI responders as opposed to patients with high initial WBC who likely went on to be TKI-nonresponders [77,78]. It has also been reported that CML patients who expressed lower levels of BCR::ABL1 transcripts 3 or/and 6 months post-treatment were significantly associated with TKI response [78,79,80]. In addition, differential gene expression panels and microRNA (miRNA) expression patterns are also useful predictive biomarkers [81,82,83,84,85]. In two recent studies, it was found that expression profiles of certain miRNAs combined with in vitro colony-forming cell assays could predict response to IM and NL [78,86]. While these are early studies, screening of endogenous biomarkers may be a valuable tool in facilitating optimal TKI selection in the future.

After the application of an appropriate front-line TKI, patients are monitored for tolerance and response. In the event of TKI toxicity to front-line IM therapy, it is recommended that the dose is first reduced before resorting to switching to a second-generation TKI [1,9]. For patients who start with second-generation TKIs that experience toxicity, it is recommended that the second-generation TKI is rotated between NL, DA, and BOS while considering the patient comorbidities [1,9]. Other options include treatment with IM or switching to a third-generation TKI at lower doses [1,9]. On the other hand, patients who develop resistance to front-line IM are recommended to switch to second-generation TKIs depending on the type of ABL1 kinase resistance mutation acquired [1,9]. If resistance continues with second-generation TKIs, then third-generation TKIs like PON and ASC are considered especially for patients harboring the T315I mutation. Although there are currently no clinical studies directly comparing PON with ASC, PON is more favored based on the results of two separate clinical studies on PON and ASC with similar study conditions [87,88,89]. Currently, several ongoing clinical trials have been applied to use second- or third-generation TKIs in combination with BCL-2 inhibitors or other agents to improve the efficacy of treatment to overcome TKI resistance.

### 4.2. Clinical Trials of Developing TKI Treatment Strategies for CML 

The approval of several major TKIs phase III clinical trials, including the comparison of efficacy of different TKIs in the treatment of CML patients, have demonstrated promising clinical results, and these clinical trials include the IRIS (IM), DASISON (DA vs. IM), ENESTnd (NL vs. IM), BFORE (BOS vs. IM), EPIC (PON vs. IM), and ASCEMBL (ASC vs. BOS) trials. Additionally, several non-BCR::ABL1 targeting compounds which include histone deacetylase inhibitors and JAK/STAT, Hedgehog, Aurora kinase, and PPAR-γ signaling pathway inhibitors have been observed to be effective against CML in phase I/II clinical trials [90]. The efficacy of these compounds are attributed to the dependency of BCR::ABL1-independent signaling mechanisms, particularly for advanced-stage CML and LSCs, and identification of novel targetable factors in CML that confer BCR::ABL1-independent resistance continues to remain a priority [47,91]. As the results of these clinical trials have already been summarized in previous publications, the rest of this review will instead highlight updates to existing and emerging phase II/III clinical trials of TKI treatment strategies [1,87,90,92].

#### 4.2.1. ASC4FIRST

Currently, ASC is approved as a third-line therapy for CML. The ASC4FIRST (NCT4971226) is an ongoing phase III multi-center clinical trial comparing ASC against an investigator-selected approved TKI as first-line therapies in newly diagnosed BCR::ABL1^+^ CML-CP adult patients (Table 1) [93,94]. Approximately 404 randomized patients will either receive ASC 80 mg once daily (QD) or one of IM (400 mg QD), NL (300 mg twice per day), DA (100 mg QD), or BOS (400 mg QD) (Table 1) [93,94]. The ASC4FIRST study is estimated to be completed in 2028 and will aim to assess major molecular response (MMR, BCR::ABL1/ABL ratio of ≤0.1% BCR::ABL1^IS^ on the international scale), adverse events, pharmacokinetics, and patient-reported outcomes of ASC vs. investigator-selected TKIs at week 48 and week 96 [93,94]. Ultimately, the study is expected to provide valuable insight into further development of first-line treatment strategies in CML.

#### 4.2.2. ASC2ESCALATE

Another ongoing clinical trial with ASC is the ASC2ESCALATE (NCT05384587), a phase II multi-center, single-arm, dose-escalation study that aims to evaluate the efficacy and safety of ASC dose-escalation in CML-CP patients as a second-line therapy (Table 1) [96]. The patients recruited for this study include a cohort of 92 adult CML-CP patients without the T315I mutation who displayed resistance or intolerance to first-line treatment (Table 1) [96]. Additionally, a cohort of 60–90 newly diagnosed CML-CP patients will also be included in this study to compare the effects of ASC as a first-line therapy [96]. Patients will receive ASC 80 mg QD, and the study will measure the proportion of patients that will achieve MMR at 12 months (Table 1) [96]. For those that do not meet response milestones, the study will also assess whether dose escalation to 200 mg QD and 200 mg BID will facilitate the achievement of MMR [96].

#### 4.2.3. PACE and OPTIC

The PACE (NCT01207440) clinical trial was a phase II trial evaluating the efficacy of single-dose PON at 45 mg QD in 449 adult CML or Ph^+^ ALL patients who were resistant or intolerant to DA or NL or harbored the T315I mutation. The results from this trial found that PON helped 60% of patients achieve a major cytogenic response (MCyR), 40% achieve MMR, and 24% achieve a 4.5-log molecular response (≤0.0032% BCR::ABL1^IS^) (Table 1) [97]. Most strikingly, the majority of these patients sustained these responses even after 40 months following dose reduction and demonstrated a 5-year overall survival rate of 73% [97]. However, significant adverse events such as cardiovascular toxicities were associated with PON treatment which led to its temporary withdrawal from the market in 2012 [97].

To adjust against the adverse effects of PON on CML patients, the OPTIC (NCT02467270) clinical trial was performed. OPTIC is an ongoing phase II randomized, dose-optimization study of PON designed to evaluate the efficacy of three different starting doses of PON in CML-CP patients who experienced resistance to more than two previous TKIs or who possess the T315I mutation [98,99]. In this study, 283 patients received either 45 mg, 30 mg, or 15 mg QD of PON and were assessed for a primary endpoint of MR^2^ (≤1% BCR::ABL1^IS^) at 12 months (Table 1) [98,99]. Primary analysis showed that 44% of patients receiving 45 mg, 29% of patients receiving 30 mg, and 23% of patients receiving 15 mg achieved MR^2^ at 12 months (Table 1) [98]. A three-year follow-up analysis of the OPTIC trial showed that the percentage of patients who achieved the primary endpoint rose to 60% in the 45 mg cohort and 40% in both the 30 mg and 15 mg cohorts (Table 1) [100]. At both 12-month and 36-month time points, the 45 mg QD dose strategy with a reduction to 15 mg QD upon reaching MR^2^ displayed the most optimal benefit-to-risk ratio [98,100]. Overall, a comparison study found that patients from the OPTIC trial had lower incidences of adverse effects compared to patients from the PACE trial and highlighted the benefits of dose optimization strategies for TKIs [101].

#### 4.2.4. Decitabine, Venetoclax, and Ponatinib for the Treatment of Philadelphia Chromosome-Positive Acute Myeloid Leukemia or Myeloid Blast Phase or Accelerated Phase CML (M.D. Anderson Cancer Center)

It has previously been reported that in a phase I/II clinical study, DA could be combined with the hypomethylating agent, decitabine, to effectively treat advanced phase CML (CML-AP, CML-BP) [102]. Furthermore, it has been shown that the anti-apoptotic BCL-2 family proteins are upregulated in CML blast crisis cells and LSCs via BCR::ABL1 signaling, and preclinical studies combining TKIs with the BCL-2 inhibitor venetoclax revealed synergistic anti-leukemic effects in BCR::ABL1^+^ myeloid leukemias [103,104]. Following up on these results, a phase II clinical trial (NCT04188405) combining PON (45 mg QD) with decitabine (20 mg/m^2^ QD), and venetoclax (400 mg QD) is currently underway to determine the efficacy of this combination in newly diagnosed or relapsed/refractory CML-AP or CML-BP (Table 1). While the trial continues to accumulate patients, recent updates have shown that out of 14 advanced-phase CML patients treated so far, 11 patients responded well with 3 experiencing continued response without stem cell therapy (Table 1) [105].

#### 4.2.5. TFR and TKI Discontinuation Trials

As discussed earlier, TFR is the desired outcome for CML, and TKI discontinuation is eligible for patients who experience a sustained DMR. The first TKI discontinuation trial was the STIM1 trial which assessed outcomes in 100 CML patients with DMR for at least 2 years who discontinued IM therapy [106]. Long-term follow up of the STIM1 trial found that 38% of patients maintained molecular remission after a median follow-up of 77 months after TKI discontinuation and that no patients experienced disease progression [106]. Another large TKI discontinuation trial called EURO-SKI followed 755 patients, which agreed with the findings from the STIM1 trial and helped establish the criteria that patient candidates for TKI stopping should have experienced at least 3 years of TKI therapy and at least 1 year of sustained DMR [107].

However, about 50% of patients who discontinue therapy experience relapse though they are able to regain DMR status with TKI re-initiation [108,109,110]. Thus, there are clinical trials that aim to determine the feasibility of patients to achieve a second TFR. The first study to investigate this was the French Nilo Post-STIM (NCT01774630) study which studied 31 patients who had relapsed after attempting TFR following IM discontinuation (Table 1) [111]. In this study, patients were treated with NL at 300 mg twice per day over two years, and it was found that TFR rates after NL discontinuation were 59% at 24 months and 42% at 48 months which were similar to the rates after the first TFR attempt (Table 1) [111]. Despite these promising results, the investigators conceded that switching from IM to NL for a second TFR was not recommended due to a relatively small sample size and a relatively high 23% rate of NL-associated adverse events [111].

Another TKI discontinuation trial is the DAstop2 (NCT03573596) clinical trial which is an ongoing study that aims to investigate TFR rates following a 2nd TFR attempt after DA treatment in patients who previously failed TFR with any TKI [112]. A total of 94 patients were enrolled in the trial and treated with DA 100 mg or 70 mg QD for two years. These patients were cleared for a second TFR attempt if patients were able to re-achieve a sustained 4-log molecular response (≤0.01% BCR::ABL1^IS^) (Table 1) [112]. Although the trial is ongoing, interim results as of 2024 showed that 66% of patients had attempted a second TFR attempt with TFR rates of 56% and 46% at 12 months and 24 months, respectively (Table 1) [112]. Additionally, 87% of patients re-achieved a 4-log molecular response within 3 months of TKI re-commencement [112]. Thus, attempting a second TFR using second-generation TKIs appears to show promising results and may inform future strategies to achieve TFR in patients.

#### 4.2.6. Olverembatinib

Olverembatinib is a new third-generation TKI that was developed to fill a need in countries where PON and ASC are not available and was approved in China for adult CML-CP or CML-AP patients who are TKI-resistant and have the T315I mutation [9]. Mechanistically, olverembatinib can bind to the ATP-binding site of both phosphorylated and unphosphorylated forms of wild-type and mutated BCR::ABL1 including the T315I mutant through extensive hydrogen bond networks [113]. Compared to the other TKIs discussed previously, olverembatinib exhibits impressive anti-leukemic activity against virtually all ABL1 kinase domain mutations and other ABL1 region compound mutations that cause TKI resistance [9].

In a multi-center, randomized phase II study (NCT04126681) in China conducted on CML-CP patients intolerant to IM, DA, and NL, the olverembatinib treatment arm significantly increased event-free survival and response rates compared to the best available therapy arm (Table 1) [114]. Two other phase II clinical trials on olverembatinib monotherapy for CML-CP (NCT03883087) and CML-AP (NCT03883100) patients in China with the T315I mutation also demonstrated favorable response rates (Table 1) [115]. Although not yet approved by the FDA for treatment in the U.S., the FDA designates olverembatinib as an orphan drug, as the drug has undergone several emerging clinical trials which highlight its strong potential as a CML therapeutic option [116].

Clinical studies with olverembatinib showed that the maximum tolerated dose was 50 mg QD and that the drug consistently achieved ≥70% complete cytogenic response (CCyR, ≤1% BCR::ABL1^IS^) and ≥50% MMR with minimal adverse effects in patients with the T315I mutation as well as those with resistance to other TKIs such as PON [117,118,119,120]. Results from a recent phase I clinical trial (NCT04260022) on 76 CML-CP and Ph^+^ ALL patients outside of China treated with olverembatinib also demonstrated promising results comparable to those acquired from Chinese patients (Table 1). In this trial, olverembatinib was able to induce overall CCyR and MMR rates of 57% and 43%, respectively in all CML-CP patients (Table 1) [121]. Strikingly, olverembatinib was also able to induce impressive CCyR and MMR rates in CML-CP patients with the T315I mutation as well as in patients who had previously failed PON and/or ASC treatment (Table 1). CCyR and MMR rates in PON-resistant patients were 53% and 38%, respectively (Table 1) [121]. For ASC-resistant patients, CCyR was 43% and MMR was 38% (Table 1) [121]. For patients that experienced prior failure to both ASC and PON, 25% were able to achieve MMR with olverembatinib (Table 1) [121]. Although this study is continuing to recruit patients, these updated results suggest that olverembatinib monotherapy was able to overcome the T315I mutation and could benefit patients who were resistant to PON and ASC [121].

A phase II study (NCT05594784) on 31 Ph^+^ ALL has also demonstrated that olverembatinib combined with venetoclax can deliver high response rates in patients with relapsed or refractory Ph^+^ ALL where the p190 BCR::ABL1 isoform is the most dominant (Table 1) [122]. Aside from targeting BCR::ABL1 in CML and Ph^+^ ALL, olverembatinib has also shown efficacy against FLT3-ITD AML and IM-resistant gastrointestinal stromal tumors by being able to inhibit several other kinases such as KIT and FLT3 among others [123]. Given that olverembatinib can overcome a broad range of TKI-resistant mutants including T315I, olverembatinib is anticipated to be an effective third-line treatment option. Regimens for using olverembatinib as a frontline treatment are also currently being investigated.

#### 4.2.7. Preclinical Developments of CML Therapeutic Strategies

There have also been promising developments in therapeutic strategies being explored on the preclinical front. Several non-BCR::ABL1 targeted therapies are known to be effective in the treatment of CML, especially when combined with TKIs, and have already been reviewed [90,124]. More recently, it has been shown that the induction of nitric oxide synthase (NOS) expression in leukemic cells leads to apoptosis and pyroptosis through various mechanisms including caspase activation, mitochondrial dysfunction, and reactive nitrogen species (RONS)-mediated modulation of the PI3K/Akt, p38MAPK/Erk1/2, and JNK pathways [125]. Exploiting NOS vulnerability in CML may thus be beneficial for therapeutic purposes and warrants further study, particularly with clinical samples.

Given that TKI therapies are rarely curative, there has also been a growing number of preclinical studies on the application of CRISPR/Cas9 in the context of CML in the last few years. Conventionally, allogenic hematopoietic stem cell (HSC) transplantation is considered the only curative treatment for CML but is associated with major issues such as graft-versus-host-disease and donor compatibility. However, these risks may be addressed with autologous transplantation of gene-edited HSCs. The CRISPR/Cas9 system has been demonstrated to effectively disrupt BCR::ABL1 at the genomic level leading to leukemia cell death in vitro and in vivo with no off-target effects [126,127,128,129]. Despite high gene-editing efficiency using the CRISPR/Cas9 system, a major barrier to curative gene therapy in CML is the presence of unedited cells that have the potential to initiate relapse after transplantation [127]. To overcome this, a proof-of-concept study described the efficacy of a CRISPR-gene trapping strategy that would allow for CRISPR-edited cells to be selected before transplantation [127]. In this study, a donor gene trap cassette carrying a fluorescent reporter gene was inserted into the BCR::ABL1 fusion site in K562 cells [127]. The selected cells were indeed shown to have abrogated BCR::ABL1 expression and were also shown to have reduced proliferation capacity and increased apoptosis in vitro and in vivo xenograft mouse models [127]. Integration of CRISPR/Cas9 with artificial intelligence design algorithms is also being explored and may lead to more efficient development of robust gene therapies against BCR::ABL1-dependent and BCR::ABL1-independent resistance mechanisms in the future [130,131]. While further studies are needed to assess the efficacy and safety of CRISPR/Cas9 gene therapy for CML in the clinical setting, the recent FDA approval of gene therapies for sickle cell anemia indicates that this is a promising avenue for the future of CML treatments.

## 5. Conclusions

The discovery of IM and the development of next-generation TKIs have revolutionized CML treatments, turning a previously fatal disease into a manageable chronic condition. The groundbreaking success in CML treatment serves as a paradigm for targeted therapies and promotes the investigation of small-molecule inhibitors in other types of malignancies. However, despite the outstanding clinical outcomes in CML patients, the disease relapses, and the emergence of resistance seems to be inevitable. Although second-generation TKIs are more potent and can target most of the IM-resistant mutants, several mutations such as the notorious T351I are still immune to most treatments and confer a poor prognosis for such patients. While approved third-generation TKIs like PON and ASC have demonstrated efficacy against the T315I mutant, some patients continue to respond sub-optimally due to their inability to overcome compound mutations. Additionally, the different p190, p210, and p230 isoforms of BCR::ABL1 may also respond differently to therapies. Therefore, targeting BCR::ABL1 alone may not be sufficient. Further deciphering the molecular and cellular network of CML pathogenesis by advanced global sequencing, proteomics, bioinformatics analysis, and artificial intelligence technology may identify key molecular targets that are playing important roles in drug resistance, disease maintenance, and progression, which will lead to the development of new combination therapies. Lastly, exploring new clinical strategies that aim to optimize TKI dose and treatment regimens may offer a fresh outlook on achieving TFR with existing TKIs that can minimize adverse events, improve quality of life, and reduce healthcare costs.

## Figures and Tables

**Figure 1 ijms-25-03307-f001:**
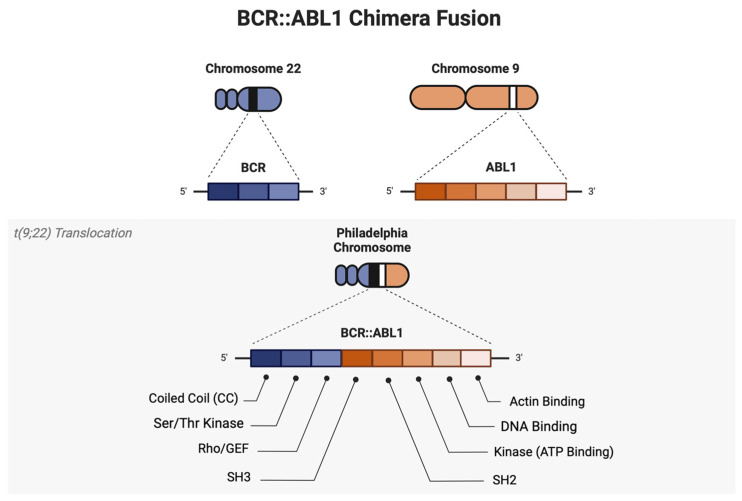
Generation of the BCR::ABL1 Fusion Protein. The BCR::ABL1 chimeric gene is formed when the ABL1 gene on chromosome 9 is translocated to the BCR gene on chromosome 22. This results in a constitutively active fusion protein that phosphorylates several downstream oncogenic signaling pathways that contribute to leukemia pathogenesis. Created with BioRender.com.

**Figure 2 ijms-25-03307-f002:**
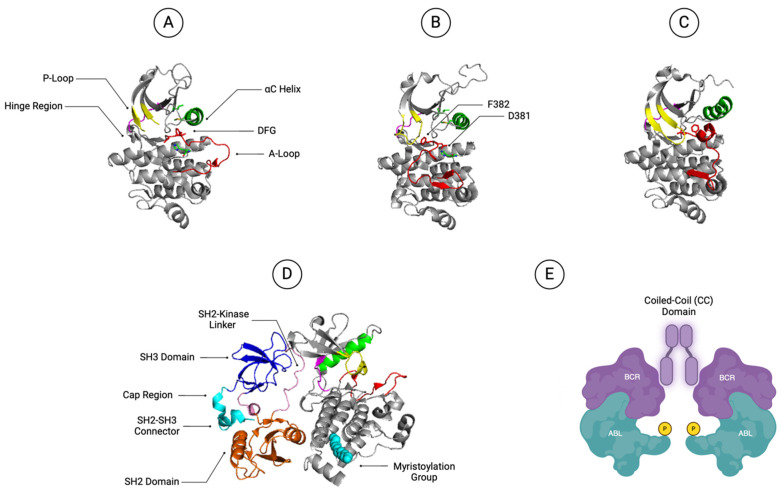
The different conformations of the ABL1 kinase domain and the ABL1 conformational changes between active and inactive states. (**A**) The ABL1 active conformation with the P-loop is represented in yellow, the A-loop in red, and the αC helix in green. (**B**) The ABL1 inactive conformation with the DFG side chains of D381 and F382 are presented in red. (**C**) The Src-like inactive conformation. (**D**) The ABL1 assembled conformation. The SH2 domain, the connector, and the SH2-kinase linker are colored in pink. The N-terminal myristoylation group and the Cap region are colored in cyan. (**E**) Tetramerization of BCR::ABL1 via the coiled-coil (CC) domain leads to proximity-induced trans-phosphorylation, depicted in yellow. Created with BioRender.com.

**Figure 3 ijms-25-03307-f003:**
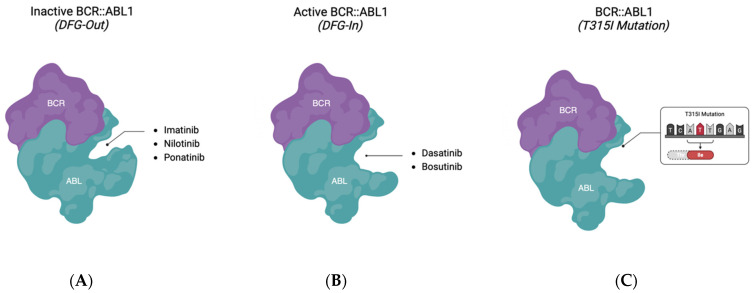
BCR::ABL1 conformation changes between active and inactive states targeted by TKIs. IM, NL, and PON target the inactive state (**A**) with the DFG motif out of the catalytic cleft, whereas DA and BOS target the active (DFG-in) state (**B**). The T315I mutation (**C**) prevents conventional TKIs from binding. Created with BioRender.com.

**Figure 4 ijms-25-03307-f004:**
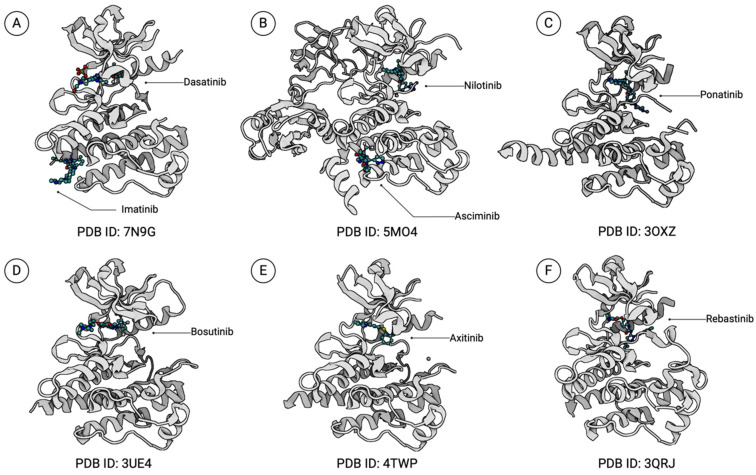
BCR::ABL1 and TKIs binding mechanisms. IM (**A**), NL (**B**), and PON (**C**) bind BCR::ABL1 in the DFG-out (inactive) state. IM relies on hydrogen bonding and van der Waals interactions, NL exhibits increased potency through extensive van der Waals contacts, and PON employs an ethynyl linker, successfully overcoming the challenging T315I mutation. DA (**A**) and BOS (**D**) bind BCR::ABL1 in the DFG-in (active) state. DA forms numerous van der Waals interactions with the hinge region of the ABL1 kinase and hydrogen bonds with the T315 residue making it susceptible to the T315I mutation. BOS binds similarly to DA but forms crucial van der Waals interactions with the pivotal T315 residue rather than hydrogen bonds. ASC (**B**) inhibits BCR::ABL1 through allosteric inhibition by targeting the myristoyl pocket. Axitinib (**E**) and rebastinib (**F**) both specifically target the BCR::ABL1 T315I mutation. Axitinib binds BCR::ABL1 in an active formation whereas rebastinib (**F**) is a non-competitive conformational switch inhibitor. Structures were obtained from the Protein Databank [44]. Created with BioRender.com.

**Table 1 ijms-25-03307-t001:** Summary of emerging TKI clinical trials for the treatment of CML. MHR, major hematologic response includes complete hematologic response (CHR) and no evidence of leukemia. CR, complete remission; CRi, CR with incomplete hematologic recovery; MLFS, morphologic leukemia-free state. MCyR, major cytogenic response; CCyR, complete cytogenic response; MR^2^ (≤1% BCR::ABL1^IS^); MR^3^/MMR (≤0.1% BCR::ABL1^IS^); MR^4^ (≤0.01% BCR::ABL1^IS^); MR^4.5^ (≤0.0032% BCR::ABL1^IS^). All response criteria are defined according to the European Leukemia-net (ELN) guidelines [3,95].

Trial Name (Clinical Trial Number)	Compound(s)	Combination	Rationale	Phase	Number of Patients	Outcome	Status
ASC4FIRST (NCT04971226)	Asciminib (80 mg QD)	Monotherapy	Comparison of efficacy of ASC as a first-line therapy against other first-line TKI:	III		TBD	Active Not recruiting
			IM 400 mg QD		404		
			NL 300 mg BID				
			DA 100 mg QD		Newly diagnosed CML-CP		
			BOS 400 mg QD				
ASC2ESCALATE (NCT05384587)	Asciminib	Monotherapy	Safety and efficacy of ASC dose escalation	II	92	TBD	Recruiting
	(80 mg, 200 mg QD, 200 mg BID)				CML-CP with prior TKI failure		
PACE (NCT01207440)	Ponatinib (45 mg QD)	Monotherapy	Safety and efficacy of PON to overcome the T315I mutation	II	449	CML-CP (267 patients):	Completed
						60% MCyR	
						54% CCyR	
					CML-CP, AP, BP	40% MMR	
					Ph+ ALL	24% MR^4.5^	
					Resistant to DA, NL or have the T315I mutation	CML-AP (83 patients):	
						61% MHR	
						49% MCyR	
						31% CCyR	
						22% MMR	
						CML-BP (62 patients):	
						31% MHR	
						23% MCyR	
						38% CCyR	
						13% MMR	
						Ph+ ALL (32):	
						47% MCyR	
						38% CCyR	
OPTIC (NCT02467270)	Ponatinib (45 mg, 30 mg, 15 mg QD)	Monotherapy	Safety and efficacy of 3 different starting doses of PON	II	283	Patients with T315I MR^2^	Active Not recruiting
						64% (45 mg)	
						25% (30 mg)	
						16% (15 mg)	
						Patients without T315I MR^2^	
					CML-CP with prior TKI failure or have the T315I mutation	59% (45 mg)	
						44% (30 mg)	
						46% (15 mg)	
DAC-VEN-PON (NCT04188405)	Ponatinib (45 mg QD)	+Decitabine (20 mg/m^2^ QD) +Venetoclax (400 mg QD)	Safety and efficacy of DAC-VEN-PON combination Assess BCL-2 dependency in response to treatment regimen	II	14	11 patients responded	Active Not recruiting
					CML-AP, BP with prior TKI or chemotherapy exposure	40% CR/CRi	
						33% MLFS	
Nilo Post-STIM (NCT01774630)	Nilotinib (300 mg BID)	Monotherapy	Safety and efficacy of NL to achieve 2nd TFR after prior IM discontinuation	II	31	7 patients discontinued therapy after experiencing adverse events	Completed
					CML patients with molecular relapse after IM discontinuation attempt	22 patients achieved TFR rates of:	
						59.1% (12 months 95% CI: 41.7–83.7%) 42.1% (24 months 95% CI: 25–71%)	
DAstop2 (NCT03573596)	Dasatinib (100 mg, 70 mg QD)	Monotherapy	Safety and efficacy of DA to achieve 2nd TFR after prior failed TKI discontinuation attempt	II	94	62 patients attempted 2nd TFR attempt with TFR rates of:	Recruiting
					CML-CP patients who relapsed after 3 years of TKI therapy and achieved deep molecular response (EURO-SKI)	61% (6 months) 56% (12 months) 46% (24 months)	
HQP1351 vs. BAT (NCT04126681)	Olverembatinib (40 mg QD)	Monotherapy	Safety and efficacy of olverembatinib compared to best available therapy (BAT)	II	144	97 patients discontinued treatment due to adverse events	Active Not recruiting
						Olverembatinib arm:	
					CML-CP patients resistant and/or intolerant to IM, DA, and NL	85% CHR	
						48% MCyR	
						36% CCyR	
						27% MMR	
						BAT arm:	
						35% CR	
						30% MCyR	
						16% CCyR	
						8% MMR	
HQP1351-CP (NCT03883087)	Olverembatinib (40 mg QD)	Monotherapy	Safety and efficacy of olverembatinib against CML-CP	I/II	127	79% MCyR (95% CI: 70–85%)	Active Not recruiting
						69% CCyR (95% CI: 60–77%)	
					CML-CP patients with the T315I mutation	56% MMR (95% CI: 47–64%)	
						44% MR^4.0^ (95% CI: 35–52%)	
						39% MR^4.5^ (95% CI: 47–64%)	
HQP1351-AP (NCT03883100)	Olverembatinib (40 mg QD)	Monotherapy	Safety and efficacy of olverembatinib against CML-AP	II	38	47% MCyR (95% CI: 31–62%)	Active Not recruiting
						47% CCyR (95% CI: 31–62%)	
					CML-AP patients with the T315I mutation	45% MMR (95% CI: 28–60%)	
						39% MR^4.0^ (95% CI: 22–56%)	
						32% MR^4.5^ (95% CI: 18–48%)	
HQP1351-CML/Ph^+^ ALL (NCT04260022)	Olverembatinib (30 mg, 40 mg, 50 mg QD)	Monotherapy +Blinatumomab	Safety and efficacy of olverembatinib against CML-CP, AP and BP or Ph^+^ ALL in patients resistant to PON and ASC	I	57 CML-CP 19 Ph^+^ ALL	12 CML-CP and 7 Ph^+^ ALL patients discontinued treatment due to adverse events	Recruiting
						CML-CP:	
						57% CCyR	
						43% MMR	
						CML-CP patients with ≥4 TKI failures:	
						57% CCyR	
						42% MMR	
						CML-CP with T315I:	
					Previously treated and resistant to PON and/or ASC	60% CCyR	
						44% MMR	
						CML-CP with prior PON failure:	
						53% CCyR	
						38% MMR	
						CML-CP with prior ASC failure:	
						43% CCyR	
						38% MMR	
						CML-CP with prior PON and ASC failure:	
						25% MMR	
						Ph^+^ ALL:	
						23% MMR	
HQP1351-VEN (NCT05594784)	Olverembatinib (40 mg QD)	+Venetoclax (100 mg d1, 200 mg d2, 400 mg d3–d28) +Chemotherapy (10 mg methotrexate, 50 mg cytarabine, 10 mg dexamethasone)	Safety and efficacy of olverembatinib combined with venetoclax and chemotherapy against Ph^+^ ALL	II	31 Ph^+^ ALL patients with relapsed or refractory disease	32% (10) MMR 6.5% (2) < MMR 61% (19) MR^4.5^	Recruiting

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
