# Peer review of "Clinical Insights into Structure, Regulation, and Targeting of ABL Kinases in Human Leukemia"

_ijms, 2024, doi:10.3390/ijms25063307_

Round 1

Reviewer 1 Report

Comments and Suggestions for Authors

In the current review, the authors have extensively reviewed Critical regions of BCR:ABL1 in terms of their structural and molecular properties. They have updated the knowledge of the crucial function of BCR::ABL1 kinase activity in building a strong foundation for the efficient development of CML molecularly targeted therapy. This article also covers a thorough discussion of current combination treatment options and comparisons of responses and resistance to several BCR:ABL1 TKIs in clinical research.

I don’t have a lot of concern in terms of description. The author can detail the different TKIs by making separate sections for each TKIs like Imatinib, Dasatinib, Nilotinib etc.

They can discuss a little bit about latest preclinical studies in CML in one paragraph.

Also, they can talk about the level of biochemical metabolites like nitrate in CML and how modulating its level and can be beneficial or deregulatory for the disease in preclinical and clinical studies. Specifically, they can talk about recent articles published around 2019 to 2021 where it has been shown that Nitric oxide have lead to differentiation of neutrophils in CML and induction of iNOS and nNOS leads pyroptotic and apoptotic cell death

Author Response

Please see the attached response letter.

Reviewer 2 Report

Comments and Suggestions for Authors

The review article “Clinical Insights Into Structure, Regulation and Targeting of ABL Kinases in Human Leukemia” summarized the TKIs of ABL1 for CML including each structural features and current treatment options including stop TKI strategy and ongoing clinical trials. This review article was informative, especially structure and mode of action, but the volume of clinical data was relatively small. Considering limited word number of the article, the reducing volume of clinical data might be reasonable. However, I believe that the valuable table and figure can provide a lot of information without a lot of words. The abstract revealed that “comparison of response and resistance to multi TKIs” but the authors cut these points in main document. Therefore, I recommend that the authors should add the table for several pivotal trials and ongoing clinical trials including stopping TKIs and new combination therapies.

Author Response

PLease find the attached response letter.

Reviewer 3 Report

Comments and Suggestions for Authors

Summary: The manuscript provides a comprehensive review of the structural and molecular properties of the BCR:ABL1 fusion protein in CML and the role of ABL kinases in promoting cellular survival. The paper discusses the development of TKIs as a successful therapeutic approach in CML treatment while also addressing the challenges of disease relapses and the emergence of resistant clones. The comparison of responses to various BCR:ABL1 TKIs and the exploration of combination treatment strategies are key strengths of this review.

General Comments: The review covers various topics related to the molecular mechanisms underlying CML and the therapeutic strategies targeting ABL kinases. The manuscript effectively highlights the importance of understanding BCR:ABL1 kinase activity in developing targeted therapies for CML. The identification of acquired mutations in the ABL1 kinase domain and the presence of quiescent CML leukemia stem cells as factors contributing to treatment resistance is particularly insightful. The references cited are relevant and contribute to the credibility of the review. However, further discussion on the potential future directions in CML treatment and the integration of emerging technologies like artificial intelligence could enhance the manuscript.

Specific Comments:

·      Introduction: Provide more literature and elaborate upon the current challenges in achieving treatment-free remission in CML patients.

·      Introduction: Consider elaborating on the BCR:ABL1 fusion protein's specific structural features contributing to its constitutive kinase activity.

·      Recommend including tables to enlist the ongoing trials evaluating novel TKIs 1) with data and 2) enrolling trials.

·      Discussion: In the discussion on the clinical trials, please include the total number of patients enrolled/ patients in each arm for phase III trials, specific outcome measures or patient responses from the trials mentioned, and corresponding p-value or confidence intervals.

·      Structural and Molecular Description of ABL1 Regulation: The discussion could be further elucidated by including specific examples of how these conformational changes impact kinase activity and downstream signaling pathways.

·      Clinical Efficacy of TKIs and New Treatment Strategies: Please elaborate on the resistance mechanisms to third-generation TKIs like Ponatinib and Asciminib, particularly in emerging mutations in the ABL1 kinase domain and alternative signaling pathways driving resistance.

  • Consider incorporating recent studies or clinical trials evaluating the efficacy of BCL-2 inhibitors in combination with TKIs in a tabular format, highlighting the rationale behind these combination approaches and their potential clinical implications.

  • Discuss the impact of treatment discontinuation on disease relapse rates and the feasibility of achieving treatment-free remission in CML patients following prolonged TKI therapy.

Overall, the manuscript provides valuable insights into the molecular basis of CML and the therapeutic implications of targeting ABL kinases. Addressing the comments and suggestions outlined above would further enhance the clarity and depth of the review for the audience of IJMS.

Comments on the Quality of English Language

Minor editing of the English language is required; consider running a proofreading tool such as Grammarly.

Author Response

Please find the attached response letter.
